# Characteristics of Early Pancreatic Cancer: Comparison between Stage 1A and Stage 1B Pancreatic Cancer in Multicenter Clinical Data Warehouse Study

**DOI:** 10.3390/cancers16050944

**Published:** 2024-02-26

**Authors:** Hyo Suk Kim, Young Hoon Choi, Jae Sin Lee, Ik Hyun Jo, Sung Woo Ko, Kyu Hyun Paik, Hyun Ho Choi, Han Hee Lee, Yeon Soo Lim, Chang Nyol Paik, In Seok Lee, Jae Hyuck Chang

**Affiliations:** 1Department of Internal Medicine, Bucheon St. Mary’s Hospital, College of Medicine, The Catholic University of Korea, Bucheon 14647, Republic of Korea; kimhyosuk@catholic.ac.kr; 2Department of Internal Medicine, Seoul St. Mary’s Hospital, College of Medicine, The Catholic University of Korea, Seoul 06591, Republic of Korea; yhchoi@catholic.ac.kr (Y.H.C.); isle@catholic.ac.kr (I.S.L.); 3Department of Internal Medicine, Incheon St. Mary’s Hospital, College of Medicine, The Catholic University of Korea, Incheon 21432, Republic of Korea; eaglelee111@catholic.ac.kr; 4Department of Internal Medicine, St. Vincent’s Hospital, College of Medicine, The Catholic University of Korea, Suwon 16247, Republic of Korea; jera0131@catholic.ac.kr (I.H.J.); cumc@catholic.ac.kr (C.N.P.); 5Department of Internal Medicine, Eunpyeong St. Mary’s Hospital, College of Medicine, The Catholic University of Korea, Seoul 03312, Republic of Korea; gogo930@catholic.ac.kr; 6Department of Internal Medicine, Daejeon St. Mary’s Hospital, College of Medicine, The Catholic University of Korea, Daejeon 34943, Republic of Korea; qhyun515@catholic.ac.kr; 7Department of Internal Medicine, Uijeongbu St. Mary’s Hospital, College of Medicine, The Catholic University of Korea, Uijeongbu 11765, Republic of Korea; chlgg@catholic.ac.kr; 8Department of Internal Medicine, Yeouido St. Mary’s Hospital, College of Medicine, The Catholic University of Korea, Seoul 07345, Republic of Korea; hanyee@catholic.ac.kr; 9Department of Radiology, Bucheon St. Mary’s Hospital, College of Medicine, The Catholic University of Korea, Bucheon 14647, Republic of Korea; yeslim@catholic.ac.kr

**Keywords:** early detection of cancer, pancreatic cancer, pancreatic intraductal neoplasm, prognosis, survival

## Abstract

**Simple Summary:**

Early detection and treatment are known to enhance the prognosis for pancreatic cancer, but there is little known about the characteristics of early pancreatic cancer. We thus aimed to identify the characteristics, clues for early detection, and prognostic factors for early pancreatic cancer by analyzing a large number of patients with stage 1 pancreatic cancers. Our study included 257 stage 1 pancreatic cancer patients and revealed that new-onset diabetes and intraductal papillary mucinous neoplasm (IPMN) were associated with early pancreatic cancers smaller than 2 cm (T1, stage 1A). In particular, pancreatic cancers smaller than 1 cm were closely associated with IPMN. These results indicate that appropriate follow-up and treatment of IPMN are important for the early detection and curing of pancreatic cancer. Since perineural invasion is a significant risk factor for both poor overall and disease-free survival in patients with stage 1 pancreatic cancer, active adjuvant therapy is required in stage 1 pancreatic cancers that feature the presence of perineural invasion.

**Abstract:**

Background: Little is known about the characteristics of early pancreatic cancer. We aimed to identify the characteristics, clues for early detection, and prognostic factors for early pancreatic cancer by analyzing a large number of patients with stage 1 pancreatic cancer. Methods: A clinical data warehouse that includes databases of all the medical records of eight academic institutions was used to select and analyze patients with pancreatic cancer that had been diagnosed from January 2010 to May 2023. Results: In total, 257 stage 1 pancreatic cancer patients were included. There were 134 men (52%), and the average age was 67.2 ± 9.9 years. Compared to patients with stage 1B pancreatic cancer (2–4 cm), patients with stage 1A pancreatic cancer (≤2 cm) had more tumors in the body and tail than in the head (*p* = 0.028), more new-onset diabetes and less old diabetes (*p* = 0.010), less jaundice (*p* = 0.020), more follow-up of IPMN (intraductal papillary mucinous neoplasm, *p* = 0.029), and more histories of acute pancreatitis (*p* = 0.013). The pathological findings showed that stage 1A pancreatic cancer involved more IPMNs (*p* < 0.001) and lower pancreatic intraepithelial neoplasia (*p* = 0.004). IPMN was present in all 13 pancreatic tumors that were smaller than 1 cm. In multivariate analysis, positive resection margin (odds ratio [OR] 1.536, *p* = 0.040), venous invasion (OR 1.710, *p* = 0.010), and perineural invasion (OR 1.968, *p* = 0.002) were found to be risk factors affecting disease-free survival, while old diabetes (odds ratio [OS] 1.981, *p* = 0.003) and perineural invasion (OR 2.270, *p* = 0.003) were found to be risk factors affecting overall survival. Conclusions: IPMN is closely associated with early pancreatic cancer and may provide an opportunity for early detection. The presence of perineural invasion was a crucial prognostic factor for both overall and disease-free survival in patients with stage 1 pancreatic cancer.

## 1. Introduction

Pancreatic cancer has the poorest prognosis among major cancers. The recently reported five-year relative survival rate of pancreatic cancer was 13% by the American Cancer Society (https://www.cancer.org/cancer/types/pancreatic-cancer/detection-diagnosis-staging/survival-rates.html, accessed on 1 February 2024) and 15.9% by the Korean Cancer Control Institute (https://cancer.go.kr/lay1/S1T648C650/contents.do, accessed on 1 February 2024). Survival rates for pancreatic cancer remain low despite ongoing advances in treatment. The early detection and treatment of low-stage pancreatic cancer is considered to be a good way to improve prognosis [1], yet the characteristics of early pancreatic cancer are still poorly understood. Pancreatic cancer is an uncommon cancer, as it affects 19.2 people in the US and 16.4 people in South Korea per 100,000 people. Stage 1 pancreatic cancer is even rarer, as it represents only 2.1% to 10.4% of all pancreatic cancers [2,3]. More specifically, stage 1A pancreatic cancers make up only 1.9% of all pancreatic cancers in the US [4]. As a result, it is difficult to study large numbers of individuals with early-stage pancreatic cancer, and there has been little research analyzing these uncommon early pancreatic cancers [2]. 

Pancreatic cancer stage can be determined according to the eighth edition TNM staging system of the American Joint Committee on Cancer (AJCC) Staging Manual [5]: T1 ≤ 2 cm, T2 > 2 cm but ≤ 4 cm, T3 > 4 cm, and T4 tumor involves the celiac axis, the superior mesenteric artery, and/or common hepatic artery; N0 no regional lymph node, N1 metastasis in 1–3 regional lymph nodes, and N2 metastasis in ≥ 4 regional lymph nodes; M0 no distant metastasis and M1 distant metastasis. The stages are 1A (T1N0M0), 1B (T2N0M0), 2 (T3 or N1), 3 (T4 or N2), and 4 (M1). Recent published survival data according to AJCC 8th edition showed that median overall survival was 19.6 months for stage 1, 14.7 months for stage 2, 14.3 months for stage 3, and 6.1 months for stage 4 [6]. In stage 1, median overall survival was 35.8 months for stage 1A and 16.8 months for stage 1B. The risk factors associated with development of pancreatic cancer include obesity [7], type 2 diabetes [8], cigarette smoking [9], family history of pancreatic cancer [10,11], and chronic pancreatitis [12]. Some pancreatic ductal adenocarcinomas arise from macroscopic cystic precursors, intraductal papillary mucinous neoplasms (IPMNs) and mucinous cystic neoplasms.

A clinical data warehouse is a platform that allows information from multiple institutions to be shared anonymously, which makes it possible to identify a large number of patients with pancreatic cancer, including those with stage 1 pancreatic cancers. Thus, we aimed to identify the characteristics of early pancreatic cancer, to provide clues for early detection and identify prognostic factors for early pancreatic cancer by analyzing a large number of patients with stage 1 pancreatic cancer using a clinical data warehouse.

## 2. Materials and Methods

### 2.1. Study Population and Data Collection

A clinical data warehouse that contains all of the medical records from eight academic institutions was used to select and study the patients. This study included patients who were diagnosed with pancreatic cancer between January 2010 and May 2023. Of the 9428 patients with pancreatic cancer who met the eligibility criteria, 257 patients (2.7%) were found to have stage 1 pancreatic cancer, as histopathologically determined by surgery according to the AJCC 8th edition. Stage 1 pancreatic cancer includes a maximum tumor diameter of 2 cm or less (T1) and a maximum tumor diameter of 2–4 cm (T2) with no regional lymph node metastasis, and no distant metastasis. 

The following clinical information was analyzed: height, weight, age, gender, serum CA 19-9 level, smoking history, drinking history, family history of cancers, history of acute and chronic pancreatitis, follow-up period, death, cause of death, survival period, recurrence time, recurrence site, neoadjuvant therapy, and adjuvant therapy. Surgical findings and pathological examination comprised the name of the surgery, tumor location, tissue diagnosis, degree of differentiation, tumor size, IPMN accompanied by cancer, lymph node metastasis, lymphatic, venous, perineural invasion, and marginal status. Lastly, imaging tests, including CT and MRI, comprised tumor size, main pancreatic duct dilatation, pancreas atrophy, and CBD (common bile duct) invasion. 

### 2.2. Definitions

Age, BMI (body mass index), CA 19-9 level, and jaundice were recorded as of the time of diagnosis. Smoking and drinking were divided according to the presence or absence thereof. In classifying pancreatic tumor locations, the head included the uncinate process, and the body included the neck. (Figure 1). Diabetes diagnosed just before pancreatic cancer was defined as new-onset diabetes. Overlapping symptoms were analyzed separately. IPMN accompanied by cancer was diagnosed by histologic findings in the surgical specimen.

### 2.3. Statistical Analysis

The results are reported in the form of frequency (percent) or mean ± standard deviation unless otherwise specified. The categorical and continuous data of the clinical characteristics in the two groups were compared by Pearson’s chi–square test/Fisher’s exact test and Student’s *t* test. The life table method was used to determine the median survival time and five-year-survival rate. The cumulative survival of pancreatic cancer was plotted using the Kaplan–Meier method and compared by Log rank test. In both univariate and multivariate analyses, Cox regression analyses were used to identify risk factors associated with pancreatic cancer patients’ survival. Multivariate analysis was performed on the factors that were found to be significant in univariate analysis. The odds ratios (ORs) and their 95% confidence intervals (CIs) are presented together. A *p* value < 0.05 was considered to indicate statistical significance. SPSS (SPSS for Windows, version 20; Chicago, IL, USA) was used for all statistical analyses.

## 3. Results

### 3.1. Characteristics of Patients with Stage 1 Pancreatic Cancer 

In total, 257 stage 1 pancreatic cancer patients were included (Table 1). There were 134 men (52%) and 123 women (48%). The average age was 67.2 ± 9.9 years, and 203 (79%) were over 60 years old. The follow-up time for stage 1B (2–4 cm) patients was 25.1 ± 22.5 months, which was shorter than the follow-up time of 33.8 ± 24.7 months for stage 1A (≤2 cm) patients (*p* = 0.011). Patients with stage 1A pancreatic cancer had more tumors in the body and tail than in the head (*p* = 0.028), more new-onset diabetes and less old diabetes (*p* = 0.010), and lower serum CA 19-9 level (*p* < 0.001) compared to patients with stage 1B pancreatic cancer. However, there were no differences in smoking, alcohol use, BMI, or family history of pancreatic cancer between patients with stage 1A and 1B. 

### 3.2. Reasons for Initial Examination and Image Findings of Stage 1 Pancreatic Cancers 

Prior to the diagnosis of pancreatic cancer, the patients visited the hospital for a variety of reasons (Table 2). Compared to stage 1B patients, stage 1A pancreatic cancer patients were more likely to have a follow-up for IPMN and have new-onset diabetes or acute pancreatitis (*p* = 0.029, 0.001, and 0.013, respectively). On the other hand, stage 1B patients experienced jaundice more frequently (*p* = 0.020). Meanwhile, abdominal pain, back pain, nausea, vomiting, diarrhea, and constipation did not differ between stage 1A and stage 1B.

As can be seen from the CT and MRI findings, main pancreatic duct dilatation was similarly observed in 45 (74%) stage 1A and 145 (75%) stage 1B cases. CBD invasion was more predominant in stage 1B than stage 1A (*p* = 0.028). However, smaller pancreatic tumors (stage 1A) were significantly associated with mural nodules and cystic size increases (*p* < 0.001 and *p* = 0.012, respectively). Some pancreatic cancers could not be observed on CT or MRI; there were 14 tumors (5%) that could not be observed on either CT and MRI, 17 tumors (10%) that could be identified on MRI but not on CT, and 11 tumors (6%) that could be identified on CT but not on MRI. However, there was no difference in invisible tumors between stage 1A and stage 1B. When the pancreatic cancer sizes in the pathology were compared to the sizes shown in CT or MRI, both CT and MRI tended to measure the tumor as smaller than the actual pancreatic cancer size, although it was not statistically significant. The average size differences between pathology and CT and between pathology and MRI were 4.24 ± 7.24 mm and 3.24 ± 6.96 mm, respectively. The CT findings showed that four tumors (1.6%) were measured from 20 to 25 mm smaller than the actual tumors, and 30 tumors (11.7%) were measured from 10 to 19 mm smaller than the actual tumors. MRI revealed that 32 tumors (12.5%) were measured more than 10 mm smaller than the actual tumors. Conversely, pancreatic tumors were measured to be 10 mm or larger than the actual tumor in only six cases (2.3%) on CT and only four cases (1.6%) on MRI. 

### 3.3. Surgical Outcomes of Stage 1 Pancreatic Cancers and Clinical Characteristics of Patients with Pancreatic Cancers Smaller Than 1 cm 

Examination of the postoperative pathologic findings yielded that stage 1B pancreatic cancers showed a more positive resection margin (*p* = 0.002), lymphatic invasion (*p* = 0.008), venous invasion (*p* = 0.001), and perineural invasion (*p* < 0.001) than stage 1A cancers (Table 3). IPMNs accompanied by cancers were more often in stage 1A (*p* < 0.001), while PanIN (pancreatic intraepithelial neoplasia) was more often in stage 1B (*p* = 0.004). Thirteen patients (5.1%) had pancreatic cancer measuring smaller than 1 cm; all of them had IPMN and an invasive carcinoma component in IPMN (Table 4). Out of these 13 patients, only one (7.7%) had abnormally high CA 19-9 levels (49.1 U/mL); the other patients’ levels were within the normal limit (<37 U/mL). Seven (53.8%) had mural nodules. Consequently, stage 1A featured less lymphatic/venous/perineural invasion and more negative resection margins than stage 1B, and IPMNs were frequently accompanied by pancreatic cancer. In particular, IPMN was present in all cases of very small pancreatic tumors measuring smaller than 1 cm.

### 3.4. Survival and Factors Related to Overall Survival and Disease-Free Survival 

Patients with stage 1 pancreatic cancer showed a five-year overall survival rate of 41% and a median overall survival time of 3.47 years. The five-year overall survival rates in patients with tumors < 1 cm, 1–2 cm, and 2–4 cm in size were 100%, 59%, and 34%, respectively. Patients with tumors 1–2 cm in size had a higher five-year overall survival rate than patients with tumors 2–4 cm in size (*p* = 0.002) (Figure 2A). The five-year disease-free survival rate was 28%, and the median disease-free survival period was 1.55 years in patients with stage 1 pancreatic cancers. The five-year disease-free survival rate in patients with tumors < 1 cm, 1–2 cm, and 2–4 cm in size were 100%, 35%, and 23%, respectively. Patients with tumors < 1 cm in size had a higher five-year disease-free survival rate than patients with tumors 1–2 cm in size (*p* = 0.017), and patients with tumors 1–2 cm in size had a higher five-year disease-free survival rate than patients with tumors 2–4 cm in size (*p* = 0.001) (Figure 2B). The results of univariate analysis showed that old diabetes, jaundice, CA 19-9 level, IPMN, and perineural invasion were all associated with overall survival (*p* < 0.05), while alcohol intake, BMI ≥ 25, jaundice, CA 19-9 level, IPMN, resection margin, lymphatic invasion, venous invasion, and perineural invasion were all associated with disease-free survival (*p* < 0.05) (Table 5). Multivariate analysis revealed that the factors related to overall survival were old diabetes (OR 1.981, 95% CI 1.268–3.093, *p* = 0.003) and perineural invasion (OR 2.270, 95% CI 1.317–3.914, *p* = 0.003), and that the factors related to disease-free survival were positive resection margin (OR 1.536, 95% CI 1.021–2.312, *p* = 0.040), venous invasion (OR 1.710, 95% CI 1.137–2.574, *p* = 0.010), and perineural invasion (OR 1.968, 95% CI 1.277–3.033, *p* = 0.002). Perineural invasion was found to be a risk factor that was associated with both overall and disease-free survival. 

## 4. Discussion

Pancreatic cancer presents a poor prognosis for several reasons. First, pancreatic cancer typically grows and spreads quickly. This makes it difficult to effectively treat, particularly if metastases have already occurred. Further, there are not many options for treating pancreatic cancer; surgery is frequently the best course of action. However, if the cancer has spread or abutted major vessels, surgery may not be an option. Even though radiation and chemotherapy are frequently used to treat cancer, their efficacy may be compromised in more advanced stages. Further, compared to other forms of cancer, pancreatic cancer cells might be more resistant to radiation and chemotherapy [14]. This resistance may make it more difficult to find a cure or long-term remission. The prognosis for individuals with pancreatic cancer may be improved by ongoing research into novel and more potent medicines, such as immunotherapies and targeted regimens [15]. However, these efforts have not yet met our expectations. Since the prognosis for advanced pancreatic cancer is very poor, early pancreatic cancer detection is crucial for improving the prognosis [16,17]. On the other hand, early-stage pancreatic cancer may not cause noticeable symptoms, and by the time symptoms appear, the disease is often advanced. Moreover, there are no generally accepted screening techniques for pancreatic cancer, unlike several other malignancies. Despite these challenges, efforts must be made to detect and treat early pancreatic cancer.

Several studies have reported the incidence and survival rates of stage 1 pancreatic cancer. The median overall survival rates for 8960 patients following surgical resection for non-metastatic pancreatic adenocarcinoma have been reported to be 38 months for stage 1A and 24 months for stage 1B, according to the Surveillance, Epidemiology and End Results database (2004–2013) [18]. Using institutional databases from four referral centers across Europe and one referral center in the US, the five-year-survival rates have been reported to 39.2% for 118 stage 1A patients and 27.6% for 144 stage 1B patients [19]. According to the Japan Pancreatic Cancer Registry, out of all pancreatic cancers, the proportions of cases in stages 0, 1A, and 1B were 1.7%, 4.1%, and 6.3%, respectively [3]. Patients with stage 0 (in situ), stage 1A, and stage 1B had five-year-survival rates of 85.8%, 68.7%, and 59.7%, respectively, while the five-year-survival rate of cases with pancreatic cancer smaller than 10 mm (TS1a) reached 80.4%, although such cancers only account for 0.8% of all pancreatic cancers. A study analyzed 200 cases of stage 0 and stage 1 pancreatic cancer 14 Japanese institutions [2]. The proportions of cases in stages 0 and I were 0.7% and 2.3%, respectively. The estimated 10 years overall survival rates for stage 0, stage 1 (TS1a), and stage 1 (TS1b, tumor size 11–20 mm) were 94.7%, 93.8%, and 78.9%, respectively. Therefore, stage 1 pancreatic cancer is uncommon, but it has a favorable prognosis. In the present study, there were only 61 (0.6%) and 196 (2.1%) patients with stage 1A and stage 1B pancreatic cancer, respectively among all pancreatic cancers, but their five-year-survival rates were high at 64% and 34%, respectively. In particular, early tumors measuring 10 mm or smaller have an extremely good survival rate. 

The goal of early detection pancreatic cancer is to enhance prognosis. Since currently available data demonstrated a difference of more than 10% in survival rates between stages 1A and 1B, it is important to detect stage 1A pancreatic cancer. Therefore, it is essential to understand and characterize stage 1A pancreatic cancer. There have been very few studies examining the clinical manifestations of early pancreatic cancer, and little is known about their characteristics. The present study discovered some clinical distinctions between pancreatic cancer measuring 2–4 cm (stage 1B) and those measuring smaller than 2 cm (stage 1A). New-onset diabetes was more frequent in stage 1A pancreatic cancer, and old diabetes was more frequent in stage 1B, suggesting that new-onset diabetes may provide a clue for detecting pancreatic cancer at earlier stages. There was also more jaundice in stage 1B. Although jaundice is generally considered to be helpful in detecting early pancreatic cancer, our study showed that jaundice is more common in larger tumors in stage 1. Therefore, jaundice was not considered to be a suitable symptom for detecting early curable cancer (stage 1A). History of acute pancreatitis and follow-up of IPMN were more common in patients with stage 1A pancreatic cancer, indicating that these conditions play a significant role in detecting early pancreatic cancer. Further, the number of cases accompanied by IPMN in the surgical specimen was significantly larger in stage 1A than in stage 1B (34% vs. 10%). Therefore, cases of stage 1 pancreatic cancer associated with IPMN could be smaller and earlier cancers.

Pancreatic cancer evaluation currently relies heavily on CT, and MRI combined with MRCP allows for a more thorough examination of the morphological alterations of the pancreatic duct and parenchyma, which helps detect cancers earlier and more effectively [20]. However, in the present study, 5% of cases of pancreatic cancer were not apparent on both CT and MRI, while 16% of cases were only detectable on either CT or MRI. This demonstrates how challenging it is to detect small tumors. All tumors that were smaller than 1 cm discovered in the present study were accompanied by IPMNs, and their invasive cancer components were not discriminated by imaging studies. The actual size of the tumor tended to be larger than that measured on CT or MRI, and the tumor size on CT tended to be smaller than that measured on MRI. Therefore, it is important to keep in mind that the pancreatic cancer might actually be larger than what appears on an MRI or CT scan.

Although stage 1A pancreatic cancer has a better prognosis than stage 1B pancreatic cancer, it is not uncommon for stage 1A pancreatic cancers to recur. The five-year-survival rate of stage 1A pancreatic cancer has been reported to be 39–68% [10]. On the other hand, the prognosis for pancreatic cancers smaller than 1 cm is excellent, and all patient who had tumors smaller than 1 cm in the present study were followed up without recurrence. Thus, to achieve complete cure without recurrence, it is necessary to detect pancreatic cancer smaller than 1 cm and then perform surgery. In the present study, all 13 patients with pancreatic cancer smaller than 1 cm were associated with IPMN. Accordingly, appropriate follow-up and surgery of IPMN is one of the ways to detect early pancreatic cancer and cure it completely. 

Perineural invasion was found to be a common factor in both overall survival and disease-free survival in our study. Perineural invasion, which is the neoplastic invasion of tumor cells into or surrounding the nerves, is a common histological feature of many other cancers. Perineural invasion is particularly common in pancreatic cancer, and it is a sign of aggressive tumor behavior and a bad prognosis [21,22]. Further, a tumor’s recurrence may be associated with the intrapancreatic perineural invasion status [23]. Perineural invasion is believed to reflect the initial stages of metastasis, and it can happen without lymphatic or vascular invasion [24,25]. Patients with perineural invasion have been reported to have an overall survival of more than two years less than those without perineural invasion [26]. Moreover, early in the development of pancreatic cancer, neuroplastic alterations such as enhanced neuronal hypertrophy and sprouting have been associated with the dissemination of cancer along with perineural invasion [27,28,29]. Perineural invasion is not uncommon in cases of early pancreatic cancer, whereas lymphatic or venous invasion is uncommon. According to our study, in stage 1A patients, lymphatic invasion occurred in 15% of cases, venous invasion occurred in 11% of cases, and perineural invasion occurred in 42% of cases, suggesting that perineural invasion may occur in early pancreatic cancer. Perineural invasion was significantly related to both overall survival and disease-free survival, which showed that it is important for prognosis. Therefore, in cases of perineural invasion, more aggressive adjuvant therapy is warranted.

There are a few limitations in this study: First, it is a retrospective study. In some cases, it is not possible to obtain all necessary data in detail due to the retrospective design. However, the clinical data warehouse that we used in this study includes all the patient’s medical records, images, laboratory, and pathology results, so it enabled us to identify patients in more detail, unlike other big data sources. Second, aside from CT and MRI, endoscopic ultrasound (EUS) is important for pancreatic cancer screening, but this was not covered in this study. Unfortunately, EUS was not performed in many cases, and EUS findings were not verified, so EUS could not be included in this study. Lastly, there were a few positive radial margins in stage 1B, including seven cases with a positive radial margin of less than 1 mm. Most American pathologists set the standard for R1 as direct invasion, while in Europe, R1 is considered to be present when the distance between the tumor and the resection margin is less than 1 mm [30]. In this study, in cases of less than 1 mm, the resection margin was often almost in contact with the tumor, so the resection margin was classified as positive according to European standards.

## 5. Conclusions

Although tremendous efforts have been made, up to this point, the prognosis for pancreatic cancer has still not changed significantly. However, early detection and treatment are considered to enhance the prognosis for pancreatic cancer, although this is challenging. Our study revealed that new-onset diabetes and IPMN were associated with early pancreatic cancers that were smaller than 2 cm. In particular, pancreatic cancers smaller than 1 cm were closely associated with IPMN. Therefore, it is important for early detection to closely follow up IPMN and perform surgery at an appropriate time. In the case of newly diagnosed diabetes and acute pancreatitis, it would be helpful to check whether pancreatic cancer is present. Active adjuvant therapy can be considered to be required in early pancreatic cancer when perineural invasion is present, since perineural invasion is related to poor overall and disease-free survival in patients with stage 1 pancreatic cancer.

## Figures and Tables

**Figure 1 cancers-16-00944-f001:**
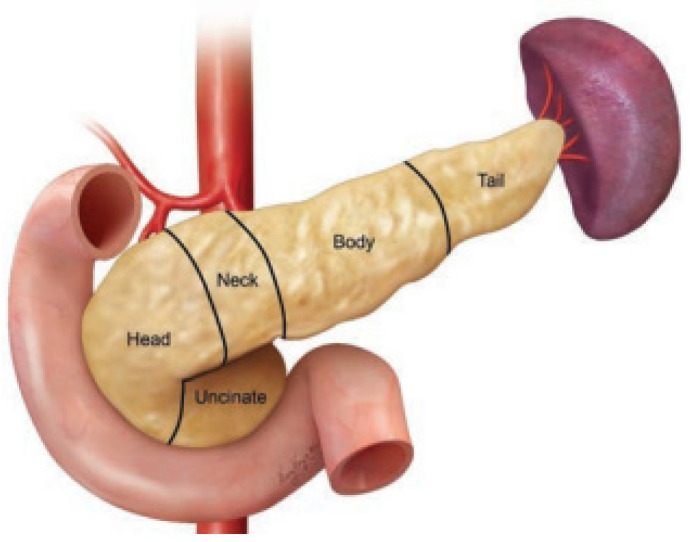
Normal pancreatic anatomy. Reprinted with permission from ref. [13]. Copyright 2020 Georg Thieme Verlag KG.

**Figure 2 cancers-16-00944-f002:**
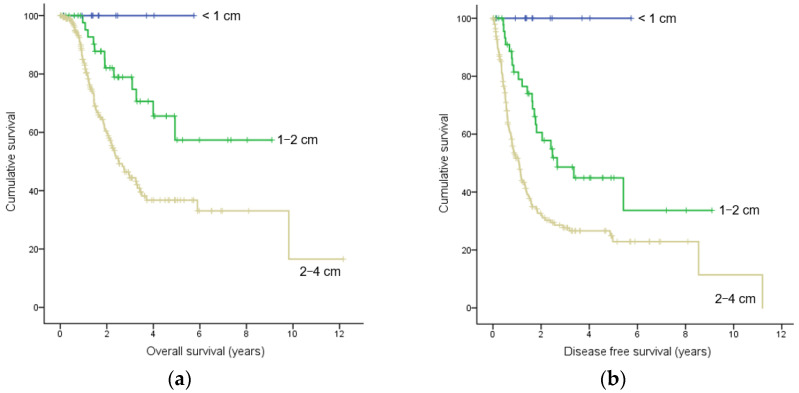
Kaplan–Meier estimates of the (**a**) overall survival and (**b**) disease-free survival according to tumor sizes. Censored subjects are indicated on the Kaplan–Meier curve as tick marks.

**Table 1 cancers-16-00944-t001:** Baseline characteristics of patients with stage 1 pancreatic cancer.

	All Cases (%), *n* = 257	Stage 1A Cases (%), *n* = 61	Stage 1B Cases (%), *n* = 196	*p*-Value
Sex				
Male	134 (52)	27 (43)	107 (55)	0.158
Female	123 (48)	34 (57)	89 (45)	
Age at diagnosis,				
mean ± SD (years)	67.2 ± 9.9	65.7 ± 11.1	67.7 ± 9.5	0.174
<50	14 (5)	6 (10)	8 (4)	0.270
50–59	41 (16)	10 (16)	31 (16)	
60–69	87 (34)	21 (34)	66 (34)	
70–79	94 (36)	17 (28)	76 (39)	
≥80	22 (9)	7 (11)	15 (8)	
Follow-up period, mo	27.1 ± 23.2	33.8 ± 24.7	25.1 ± 22.5	0.011
(mean ± SD)				
Location				0.028
Head	151 (59)	28 (46)	123 (63)	
Body	65 (25)	23 (38)	42 (21)	
Tail	41 (16)	10 (16)	31 (16)	
Risk factors				
Smoking ^†^	173 (67)	41 (67)	132 (67)	0.984
Alcohol	78 (30)	16 (26)	62 (32)	0.423
BMI, mean ± SD	23.3 ± 3.3	23.3 ± 3.3	23.3 ± 3.3	0.988
Obesity (BMI ≥ 25)	61 (24)	14 (23)	47 (24)	0.869
Familial history of pancreatic cancer	11 (4)	3 (5)	8 (4)	0.726
Diabetes				
old	91 (35)	14 (23)	76 (39)	0.010
new	14 (5)	7 (11)	7 (4)	
CA 19-9, mean ± SD	184 ± 428	95 ± 182	253 ± 475	<0.001
(U/mL)				
Neoadjunvant therapy	9 (4)	4 (7)	5 (3)	0.223

SD, standard deviation; BMI, body mass index; ^†^ current and past smokers.

**Table 2 cancers-16-00944-t002:** Reasons for initial examination in patients with stage 1 pancreatic cancer.

	All Cases (%), *n* = 257	Stage 1A Cases (%), *n* = 61	Stage 1B Cases (%), *n* = 196	*p*-Value
Symptoms	152 (59)	28 (46)	124 (63)	0.016
Abdominal pain	73 (28)	13 (21)	60 (31)	0.160
Back pain	6 (2)	1 (2)	5 (3)	1.000
Nausea/vomiting	5 (2)	3 (5)	3 (2)	0.148
Diarrhea	2 (1)	1 (2)	1 (1)	0.419
Constipation	2 (1)	1 (2)	1 (1)	0.419
Jaundice	77 (30)	11 (18)	66 (34)	0.020
Weight loss	20 (8)	8 (13)	12 (6)	0.075
Dyspepsia	11 (4)	4 (7)	7 (4)	0.297
General weakness	3 (1)	0 (0)	1 (1)	1.000
Fever	1 (0)	0 (0)	2 (1)	1.000
Headache	1 (0)	0 (0)	1 (1)	1.000
Chest pain/dyspnea	2 (1)	0 (0)	3 (2)	1.000
Routine medical check-up	12 (5)	5 (8)	7 (4)	0.135
Follow-up for other malignancy	47 (18)	8 (13)	39 (20)	0.231
Follow-up for IPMN/pancreatic cyst	12 (5)	6 (10)	6 (3)	0.029
New-onset diabetes	7 (3)	6 (10)	1 (1)	0.001
Poor sugar control in diabetes	8 (3)	1 (2)	7 (4)	0.685
Acute pancreatitis history	3 (1)	3 (5)	0 (0)	0.013

IPMN, intraductal papillary mucinous neoplasm.

**Table 3 cancers-16-00944-t003:** Surgical outcomes of stage 1 pancreatic cancers.

	All Cases (%), *n* = 257	Stage 1A Cases (%), *n* = 61	Stage 1B Cases (%), *n* = 196	*p*-Value
Surgery				0.071
PPPD	104 (40)	24 (39)	80 (41)	
Whipple	52 (20)	6 (10)	46 (23)	
DP	95 (37)	29 (48)	66 (34)	
TP	6 (2)	2 (3)	4 (2)	
Resection margin presence				
Proximal pancreatic ^†^	6 (2)	4 (7) ^†^	2 (1) ^†^	0.002 ^§^
Distal pancreatic ^‡^	4 (2)	0 (0)	4 (2)	
Radial pancreatic ^§^	38 (15)	2 (3)	36 (18) ^‡^	
Negative	209 (81)	55 (90)	154 (79)	
Lymphatic invasion				
Absent	185 (72)	52 (85)	133 (68)	0.008
Present	72 (28)	9 (15)	63 (32)	
Venous invasion				
Absent	185 (72)	54(89)	131 (67)	0.001
Present	72 (28)	7 (11)	65 (33)	
Perineural invasion				
Absent	86 (33)	33 (54)	53 (27)	<0.001
Present	171 (67)	28 (46)	143 (73)	
Pathological types				0.166
Adenocarcinoma	237 (92)	56 (92)	181 (92)	
Adenosquamous carcinoma	8 (3)	0 (0)	8 (4)	
Undifferentiated carcinoma	6 (2)	3 (5)	3 (2)	
Colloid carcinoma	6 (2)	2 (3)	4 (2)	
Accompanied by				
IPMN	41 (16)	21 (34)	20 (10)	<0.001
PanIN	55 (21)	5 (8)	50 (26)	0.004
Chronic pancreatitis	19 (7)	6 (10)	13 (7)	0.404

PPPD, pancreas-preserving pancreaticoduodenectomy; DP, distal pancreatectomy; TP, total pancreatectomy; IPMN, intraductal papillary mucinous neoplasm; PanIN, pancreatic intraepithelial neoplasia. ^†^ Including 2 cases with margin presence of PanIN-3 and 1 case with margin presence of IPMN; ^‡^ Including 1 case with margin presence of IPMN; ^§^ Including 7 cases with resection margin less than 1 mm.

**Table 4 cancers-16-00944-t004:** Characteristics of pancreatic cancer less than 1 cm in stage 1 pancreatic cancer.

	Tumor Less Than 1 cm (%), *n* = 13
Sex, male/female	5/8 (38.5/61.5)
Age at diagnosis, mean ± SD (years)	69.5 ± 11.8
Symptoms	
Abdominal/back pain	1 (7.7)/0 (0)
Nausea or vomiting/diarrhea/constipation	1 (7.7)/1 (7.7)/1 (7.7)
Jaundice/weight loss	0 (0)/0 (0)
Routine medical check-up	4 (30.8)
Follow-up for other malignancy	2 (15.4)
Follow-up for IPMN/pancreatic cyst	1 (7.7)
Poor glucose control in diabetes	0 (0)
Acute pancreatitis history	3 (23.1)
Location	
Head/body/tail	5 (38.5)/6 (46.2)/2 (15.4)
Diabetes	
Old/new	3 (23.1)/2 (15.4)
Accompanied by	
IPMN/PanIN/chronic pancreatitis	13 (100)/3 (23.1)/1 (7.7)
CA 19-9, U/mL (mean ± SD)	14.2 ± 13.1 ^†^
Imaging abnormalities	
MPD dilatation	9 (69.2)
Invisible tumor mass	10 (76.9) ^‡^
Increasing cyst size	1 (7.7)
Mural nodule	7 (53.8)
CBD invasion	0 (0)
Surgery	
PPPD/DP	5 (38.5)/8 (61.5)
Resection margin	
Proximal/distal/radial/negative	1 (7.7) ^§^/0 (0)/0 (0)/12 (92.3)
Invasion	
Lymphatic/venous/perineural	0 (0)/0 (0)/2 (15.4)
Recurrence/death	0 (0)/0 (0)

IPMN, intraductal papillary mucinous neoplasm; PanIN, pancreatic intraepithelial neoplasia; SD, standard deviation; CBD, common bile duct; SMV, superior mesenteric vein; SMA, superior mesenteric artery; PPPD, pancreas-preserving pancreaticoduodenectomy; DP, distal pancreatectomy. ^†^ CA 19-9 level in one patient (7.7%) was above the upper normal limit; ^‡^ Six patients showed only cysts without mass; ^§^ Including 1 case with margin presence of IPMN.

**Table 5 cancers-16-00944-t005:** Univariate and multivariate analysis of factors for overall survival and disease-free survival in stage 1 pancreatic cancer.

		Univariate			Multivariate	
Factors for overall survival	OR	95% CI	*p*-value	OR	95% CI	*p*-value
Age	1.011	0.990–1.032	0.321			
Smoking	1.241	0.818–1.881	0.310			
Alcohol intake	1.398	0.922–2.121	0.115			
BMI ≥ 25	0.610	0.365–1.018	0.059			
Family history of PC	0.797	0.293–2.171	0.658			
Follow-up for other cancers	0.831	0.485–1.421	0.498			
Old diabetes	1.582	1.041–2.403	0.032	1.981	1.268–3.093	0.003
New-onset diabetes	1.209	0.491–2.981	0.680			
Jaundice	1.577	1.047–2.374	0.029	1.237	0.800–1.912	0.339
CA 19-9 level	1.001	1.000–1.001	0.023	1.000	1.000–1.001	0.247
IPMN	0.446	0.207–0.963	0.040	0.892	0.395–2.014	0.783
Resection margin	1.554	0.979–2.468	0.061			
Lymphatic invasion	1.407	0.924–2.142	0.112			
Venous invasion	1.514	0.977 2.346	0.063			
Perineural invasion	2.278	1.404–3.698	0.001	2.270	1.317–3.914	0.003
Adjuvant therapy	0.700	0.457–1.071	0.100			
Factors for disease-free survival	OR	95% CI	*p*-value	OR	95% CI	*p*-value
Age	0.996	0.979–1.012	0.608			
Smoking	1.085	0.773–1.524	0.638			
Alcohol intake	1.417	1.007–1.995	0.046	1.274	0.879 1.845	0.201
BMI ≥ 25	0.637	0.424–0.958	0.030	0.674	0.436 1.040	0.075
Family history of PC	0.694	0.284–1.695	0.423			
Follow-up for other cancers	0.686	0.436–1.080	0.104			
Old diabetes	1.126	0.798–1.589	0.500			
New-onset diabetes	1.176	0.576–2.400	0.657			
Jaundice	1.583	1.134–2.210	0.007	1.357	0.937–1.966	0.106
CA 19-9 level	1.000	1.000–1.001	0.024	1.000	1.000–1.001	0.345
IPMN	0.432	0.239–0.779	0.005	0.746	0.391–1.425	0.375
Resection margin	1.659	1.130–2.438	0.010	1.536	1.021–2.312	0.040
Lymphatic invasion	1.478	1.046–2.087	0.027	0.819	0.536–1.250	0.355
Venous invasion	1.927	1.363–2.797	<0.001	1.710	1.137–2.574	0.010
Perineural invasion	2.530	1.707–3.750	<0.001	1.968	1.277–3.033	0.002
Adjuvant therapy	1.133	0.785–1.635	0.504			

OR, odds ratio; CI, confidence interval; BMI, body mass index; PC pancreatic cancer; IPMN, intraductal papillary mucinous neoplasm.

## Data Availability

The datasets generated or analyzed during the current study are not publicly available due to privacy policies but are available from the corresponding author on reasonable request.

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
