# Peer review of "Characteristics of Early Pancreatic Cancer: Comparison between Stage 1A and Stage 1B Pancreatic Cancer in Multicenter Clinical Data Warehouse Study"

_cancers, 2024, doi:10.3390/cancers16050944_

Round 1
Reviewer 1 Report
Comments and Suggestions for Authors
In the review by Kim and colleagues the authors tried to determine if there is any clinical characteristics that can be used to differentiate between or detect pancreatic cancer at early stages. Pancreatic cancer is usually detected at very late stages, at which point the disease has already metastasized to adjacent organs and the treatment options as well as response is poor. Early detection is important for better prognosis and this is lacking in pancreatic cancer. Here, the authors used medical records from a large set of datasets from 8 academic Institutions to identify characteristics, clues and prognostic factors that might help to reveal the very early stages of pancreatic cancers; it is a retrospective study but with the medical records, images, lab work and pathology results from the patients, the authors conducted a thorough analysis and determined that new-onset diabetes and IPMN are associated with early pancreatic cancers. The authors state that endoscopic ultrasound along with MRI or CT scan could provide stronger evidence for incidence of pancreatic cancer. If this knowledge helps with early detection of pancreatic cancers it will be a big help to those afflicted with the disease.
Minor comments:
Inclusion of a list of abbreviations and the expansions would be helpful.
Line 139: The authors state that the patients visited the hospital for a variety of reasons, what exactly are the reasons?
Inclusion of an image of the pancreas with the anatomy shown would help the readers understand what the authors are discussing when they mention head, tail, neck or other parts of the pancreas.
Author Response
Inclusion of a list of abbreviations and the expansions would be helpful.
Abbreviations: IPMN (intraductal papillary mucinous neoplasm), OR (odds ratio), CT (computed tomography), MRI (magnetic resonance imaging), CBD (common bile duct), BMI (body mass index), confidence interval (CI), PanIN (pancreatic intraepithelial neoplasia), PPPD (pancreas preserving pancreaticoduodenectomy), DP (distal pancreatectomy), TP (total pancreatectomy), SD (standard deviation), EUS (endoscopic ultrasound)
Line 139: The authors state that the patients visited the hospital for a variety of reasons, what exactly are the reasons?
The sentence means that there are various reasons for visiting the hospital when pancreatic cancer is first diagnosed. Table 2 shows the details.
Inclusion of an image of the pancreas with the anatomy shown would help the readers understand what the authors are discussing when they mention head, tail, neck or other parts of the pancreas..
Normal pancreatic anatomy was added to Figure 1.
Taher A.;Mujtaba B.;Ramani N.S.;Patel A.;Morani A.C. The Postoperative Pancreas Imaging. J Gastrointestinal Abdominal Radiol 2020, 3, 87-98
Reviewer 2 Report
Comments and Suggestions for Authors
This manuscript uses a multicenter clinical warehouse study to analyze the clinical indicators and prognosis of early pancreatic cancer. It is found that early pancreatic cancer less than 1 cm has a 100% five-year survival rate. It also found that diabetes seems to be a risk factor for pancreatic cancer. This study gives a direction for the early diagnosis and treatment of this extremely malignant cancer, which is quite meaningful. There are a few questions I would like the author to respond to:
1. It is recommended that the author add a more detailed section on pancreatic cancer staging and prognosis, as well as possible pancreatic cancer risk factors in the introduction.
2. From the analysis in the results, it seems that the tumor size less than 1 cm is the key to the prognosis. Is it possible that other stages of pancreatic cancer besides IA or IB will have similar results?
3. It is recommended that the author can add strategic suggestions for early detection of pancreatic cancer in the conclusion. I believe this will be an important reference for health units in various countries.
Author Response
It is recommended that the author add a more detailed section on pancreatic cancer staging and prognosis, as well as possible pancreatic cancer risk factors in the introduction.
Pancreatic cancer stage can be determined according to the eighth edition TNM staging system of the American Joint Committee (AJCC) on Cancer Staging Manual [5]: T1 ≤2 cm, T2 >2 cm but ≤4 cm, T3 >4 cm, T4 tumor involves the celiac axis, the superior mesenteric artery, and/or common hepatic artery; N0 no regional lymph node, N1 metastasis in 1-3 regional lymph nodes, N2 metastasis in ≥4 regional lymph nodes; M0 no distant metastasis, and M1 distant metastasis. The stages are 1A (T1N0M0), 1B (T2N0M0), 2 (T3 or N1), 3 (T4 or N2), and 4 (M1). Recent published survival data according to AJCC 8th edition showed that median overall survival 19.6 months for stage 1, 14.7 months for stage 2, 14.3 months for stage 3, and 6.1 months for stage 4 [6]. In stage 1, median overall survival was 35.8 months for stage 1A and 16.8 months for stage 1B. Risk factors associated with development of pancreatic cancer include obesity [7], type 2 diabetes [8], cigarette smoking [9], family history of pancreatic cancer [9,10], and chronic pancreatitis [11]. Some pancreatic ductal adenocarcinomas arise from macroscopic cystic precursors, intraductal papillary mucinous neoplasms (IPMNs) and mucinous cystic neoplasms.
From the analysis in the results, it seems that the tumor size less than 1 cm is the key to the prognosis. Is it possible that other stages of pancreatic cancer besides IA or IB will have similar results?
Yes, other stages of pancreatic cancer besides IA or IB have similar results. Stages 2, 3, and 4 pancreatic cancer have a worse prognosis than stage 1, so the difference in prognosis is more pronounced compared to pancreatic cancer less than 1 cm.
It is recommended that the author can add strategic suggestions for early detection of pancreatic cancer in the conclusion. I believe this will be an important reference for health units in various countries.
Therefore, it is important for early detection to closely follow up IPMN and perform surgery at an appropriate time. In the case of newly diagnosed diabetes and acute pancreatitis, it would be helpful to check whether pancreatic cancer is present.